# Tumor Angiogenic Inhibition Triggered Necrosis (TAITN) in Oral Cancer

**DOI:** 10.3390/cells8070761

**Published:** 2019-07-22

**Authors:** Saori Yoshida, Hotaka Kawai, Takanori Eguchi, Shintaro Sukegawa, May Wathone Oo, Chang Anqi, Kiyofumi Takabatake, Keisuke Nakano, Kuniaki Okamoto, Hitoshi Nagatsuka

**Affiliations:** 1Department of Oral Pathology and Medicine, Graduate School of Medicine, Dentistry and Pharmaceutical Sciences, Okayama University, Okayama 700-8558, Japan; 2Department of Dental Pharmacology, Graduate School of Medicine, Dentistry and Pharmaceutical Sciences, Okayama University, Okayama 700-8558, Japan; 3Advanced Research Center for Oral and Craniofacial Sciences, Graduate School of Medicine, Dentistry and Pharmaceutical Sciences, Okayama University, Okayama 700-8558, Japan; 4Division of Oral and Maxillofacial Surgery, Kagawa Prefectural Central Hospital, Takamatsu, Kagawa 760-8557, Japan; 5Department of Anatomy, Basic Medicine Science College, Harbin Medical University, Harbin 150076, China

**Keywords:** tumor blood vessel, tumor angiogenic inhibition triggered necrosis (TAITN), CXCR4 antagonist, oral squamous cell carcinoma, hypoxia

## Abstract

CXCR4 is a chemokine receptor crucial in tumor progression, although the angiogenic role of CXCR4 in oral squamous cell carcinoma (OSCC) has not been investigated. Here we show that CXCR4 is crucial for tumor angiogenesis, thereby supporting tumor survival in OSCC. Immunohistochemistry on human clinical specimens revealed that CXCR4 and a tumor vasculature marker CD34 were co-distributed in tumor vessels in human OSCC specimens. To uncover the effects of CXCR4 inhibition, we treated the OSCC-xenografted mice with AMD3100, so-called plerixafor, an antagonist of CXCR4. Notably, we found a unique pathophysiological structure defined as tumor angiogenic inhibition triggered necrosis (TAITN), which was induced by the CXCR4 antagonism. Treatment with AMD3100 increased necrotic areas with the induction of hypoxia-inducible factor-1α in the xenografted tumors, suggesting that AMD3100-induced TAITN was involved in hypoxia and ischemia. Taken together, we demonstrated that CXCR4 plays a crucial role in tumor angiogenesis required for OSCC progression, whereas TAITN induced by CXCR4 antagonism could be an effective anti-angiogenic therapeutic strategy in OSCC treatment.

## 1. Introduction

The C-X-C motif chemokine receptor (CXCR4) is expressed in various hematopoietic cells such as peripheral blood lymphocytes, unprimed T cells [1], monocytes [2], pre-B cells, and plasma cells [3,4], and is also expressed in non-hematopoietic cells such as vascular smooth muscle cells [5], endothelial cells [6,7], intestinal [8], and alveolar epithelial cells [9]. CXCR4 can be bound with the ligand, stromal cell-derived factor-1 (SDF-1), also known as C-X-C motif chemokine ligand 12 (CXCL12), and the SDF-1/CXCR4 signaling axis retains hematopoietic stem cells in the bone marrow [10]. It has been shown that CXCR4 involves proliferation, infiltration, and adhesion of hematopoietic stem cells [11,12]. CXCR4 is essential for the formation of large blood vessels that feed the gastrointestinal tract in the fetal stage [13]. Notably, it has been shown that CXCR4 was also expressed in various types of cancers including glioblastoma, liver cancer, breast cancer, and esophageal cancer [14,15,16,17]. CXCR4-positive tumors often grew faster than CXCR4-negative tumors [18,19]. It has been shown that lymph node metastasis and distant organ metastasis frequently occurred in CXCR4-positive tumor cases [15,16,17]; thus, CXCR4 is considered to be a factor of poor prognosis in cancer. It was shown that CXCR4 was expressed in 50–60% of oral squamous cell carcinoma (OSCC) clinical cases [20]. CXCR4-positive OSCC tumors metastasized to lymph nodes and to distant organs more frequently than CXCR4-negative tumor cases [20,21]. Consistently, suppression of CXCR4 reduced tumor volumes and metastases to lymph nodes and distant organs in mice transplanted with OSCC cells [22]. Despite the fact that OSCC cells express a considerably high level of CXCR4, neither proliferation nor survival of the cancer cells was inhibited by CXCR4 antagonists [20,22,23]. These studies suggest that peripheral and adjacent cells around OSCC cells might play important roles in the antitumor effects of the CXCR4 antagonist. We, therefore, hypothesized that CXCR4 antagonism could repress OSCC not by directly targeting cancer cells but by targeting peritumoral stromal cells. In the present study, we first aimed to clarify which type of cells could be positive with CXCR4 in OSCC and whereby focused on CXCR4/CD34 double-positive tumor vasculatures in clinical specimens of OSCC. CD34 was first characterized as a protein that is expressed by human hematopoietic progenitor cells and has been considered a definitive marker of angiogenesis in neoplastic lesions [24]. In the presence of pathological angiogenesis, at the sprouting tips of growing vessels, the SDF-1/CXCR4 axis was shown to play a fundamental role in endothelial cell invasion, mobilization/migration, extravasation, directional migration, homing, and cell survival [25,26,27]. We therefore examined the alteration of CD34-positive blood vessels under administration of the CXCR4 antagonist. Blood vessels generally supply oxygen and nutrients crucial for tumor growth [28,29]. Such tumor blood vessels could form a favorable tumor microenvironment for tumor growth. We, therefore, focused on the histological features of OSCC enriched in blood vessels between the tumor nests. Previous studies of CXCR4 in OSCC have focused on CXCR4 expressed on cancer cells and have not reported CXCR4-positive blood vessels in OSCC.

We here aimed to evaluate (i) the subtumor distribution of CXCR4 in human clinical specimens of OSCC and (ii) the effects of a CXCR4 antagonist AMD3100, so-called plerixafor, on inhibition of tumor angiogenesis and tumor growth in mice xenografted with OSCC cells.

## 2. Materials and Methods

### 2.1. Patient Samples

Patient-derived OSCC samples were obtained from the Department of Oral Pathology, Okayama University. This study was approved by the Ethics Committee of Okayama University Graduate School of Medicine, Dentistry and Pharmaceutical Sciences (Approval number: 1608-018). The well-differentiated type of tongue squamous cell carcinoma (10 cases) was examined in the retrospective study (Table 1). All excised tumor tissues were fixed in 4% paraformaldehyde for 12 h. Tissues were embedded in paraffin wax, and sections were prepared. 

### 2.2. Cell Line and Cell Culture

Human OSCC cell line HSC-2 was obtained from Cells Bank of Japanese Collection of Research Bioresources (JCRB) at the National Institutes of Biomedical Innovation, Health and Nutrition (NIBIOHN) (Osaka, Japan). HSC-2 cells were cultured in Dulbecco’s modified Eagle’s medium (Sigma-Aldrich, St. Louis, MO, USA) supplemented with 10% fetal calf serum (FCS) (Thermo Fisher Scientific, Waltham, MA, USA), 100 U/mL penicillin, and 100 μg/mL streptomycin (Sigma-Aldrich) at 37 °C in humidified air with 5% CO_2_. 

### 2.3. Reagent

AMD3100/plerixafor (ChemScene, Monmouth Junction, NJ, USA) (10 mg) was dissolved in 20 mL of physiological saline and stored at 4 °C before use. This concentration was the same as the concentration of the drug used in the oral cancer experiment using mice [20,22].

### 2.4. Xenograft and Administration of AMD3100 to Mice

HSC-2 cells were transplanted subcutaneously to backs and heads of 5 nude mice (BALB/c-nu/nu) at 10 × 10^5^ cells per xenograft. A week after the injection, AMD3100 (50 μg/day) was administered intraperitoneally to HSC-2-transplanted mice every day for 21 d. As a control, 5 nude mice were xenografted by the above-mentioned method and administered intraperitoneally with saline. Mice were fixed by perfusion with 10% neutral buffered formalin. Tumors were excised, fixed by immersion in 10% neutral buffered formalin, dehydrated, and embedded in paraffin. Paraffin sections were prepared for hematoxylin-eosin (HE) staining and immunohistochemistry (IHC) and observed with a light microscope. 

### 2.5. Immunohistochemistry

IHC was carried out using anti-CD34 (Histofine, Tokyo, Japan; no dilution), anti-CXCR4 (Abcam, Cambridgeshire, UK; dilution of 1:300), and anti-HIF-1α (Abcam, Cambridgeshire, UK; dilution of 1:200) antibodies. Signals were enhanced by the avidin–biotin complex method (Vectastain ABC Kit, Vector Laboratories, Burlingame, CA, USA). Detection was performed with 3,3-diaminobenzidine (DAB), and the staining results were observed with an optical microscope. Tumor tissue maximum sections were used for comparison of necrosis areas, tumor vessel structures, and HIF-1α-positive areas by using ImageJ, an image processing software [31]. For cell counts in human specimens, we counted the number of lumen structures in 5 randomly selected locations of 10 OSCC patient cases. For double-fluorescent IHC, the above-mentioned CD34 and CXCR4 antibodies were diluted with Can Get Signal (Toyobo, Osaka, Japan). Anti-mouse IgG Alexa Fluor 488 and anti-rabbit IgG Alexa Fluor 568 (Life Technologies, Carlsbad, CA, USA) were used as secondary antibodies at a dilution of 1:200. After the reactions, the specimens were stained with 0.2 g/mL of DAPI (Dojindo Laboratories, Kumamoto, Japan). The staining results were observed with a fluorescence microscope.

### 2.6. Statistical Analysis

Data were recorded and entered into an electronic database during the course of the evaluation by means of Microsoft Excel (Microsoft, Inc., Redmond, WA, USA). Means and standard deviations were used for normal data distributions. The database was transferred to JMP version 11.2 for Macintosh computers (SAS Institute Inc., Cary, NC, USA) for statistical analysis. Statistical analysis was performed to investigate whether there were significant differences between the mean values of the groups. Wilcoxon’s matched-pairs signed-rank test was used for non-normally distributed measurements, and *t*-tests were used for normally distributed measurements. The level of significance was set at *p* < 0.05.

## 3. Results

### 3.1. Investigation of CXCR4-Positive Vessels in the Stroma of Human Oral Squamous Cell Carcinoma

To investigate whether CXCR4 expression in vessels could be different between tumor and nontumor areas in OSCC clinical cases, we first defined the tumor and nontumor areas in the OSCC specimens. By HE staining, a typical morphology of squamous cell carcinoma was found to be surrounded by a subepithelial connective tissue (Figure 1a,b; tumor area). The tumor cells containing eosinophilic cytoplasm and nuclear atypia formed large and small tumor nests (Figure 1b). Abundant blood vessels and fibrous connective tissue were observed between the tumor nests (Figure 1b). 

We next examined the expression and distribution of CD34 and CXCR4 in the tumor and stroma areas. CD34-positive vascular endothelial cells forming luminal structures were found in the stroma and connective tissues (Figure 1c). Tumor stroma CD34-positive blood vessels appeared to be smaller in structure than normal, while the number of blood vessels was not different. (Figure 1d). CXCR4-positive lumen structures were found in the stroma, although CXCR4 was distributed in both tumor and stromal cells (Figure 1e). Notably, CXCR4-positive lumen structures were found in the tumor area, although not in the nontumor area (Figure 1f). CXCR4-positive vessels were significant in the tumor area more abundantly than those in nontumor areas (Figure 1g). These findings indicated that CXCR4 was selectively distributed in tumor vessels of OSCC. 

To ask whether CXCR4 and CD34 could be co-distributed in the vessels, we next performed double-fluorescent IHC. It was first confirmed that CXCR4 was distributed in both tumor cells and vessel-like structures in the stroma (Figure 2a). CD34 was distributed in endothelial cells in the stroma but not in tumors (Figure 2b). CXCR4/CD34 double-positive endothelial cells were found in the stroma (Figure 2c, arrowheads). CXCR4 was not uniformly distributed in all tumor blood vessels, but it was partially localized at the constricted and branched parts of the blood vessels. Conversely, nontumor areas with CD34-positive blood vessels were CXCR4-negative (Figure 2f, arrowheads).

These findings indicated that CXCR4-positive endothelial cells existed in OSCC stromata and prompted us to hypothesize that the CXCR4-positive blood vessels could support tumor progression.

### 3.2. A CXCR4 Antagonist AMD3100 Induced Tumor Necrosis in Oral Squamous Cell Carcinoma (OSCC)-Xenotransplanted Mice

We next asked whether a CXCR4 antagonist AMD3100 could alter the tumor status of OSCC using tumor xenograft mouse model. Macroscopically, the sizes of tumors were not different between the AMD3100 and saline groups. However, surface ulceration of tumors was notably found in the AMD3100 group but not in the control group (Figure 3a,b).

On histological examination, the transplanted HSC-2 cells formed tumor nests with keratinization and maintained the histological features of well-differentiated squamous cell carcinoma (SCC) (Figure 3c). There was neither obvious vascular invasion nor perineural invasion. These findings indicated that the characteristics of HSC-2 were maintained well. Necrotic areas were broadly found in the center and periphery of tumors in the AMD3100 group (Figure 3c). Morphologically, in the necrotic areas of the two groups, nuclei of tumor cells were lost, cell cytoplasm was swollen, and eosin-stained debris was seen (Figure 3d,e) while concentrated or broken parts of the nuclei were barely found, indicating that the main manner of cell death in the tumor was necrosis, not apoptosis. The necrotic areas were significantly enlarged in the AMD3100 group compared to the control group (Figure 3f). These findings indicated that CXCR4 antagonism induced tumor necrosis in OSCC in vivo.

### 3.3. Pathophysiological Structure of Tumor Angiogenic Inhibition with Tumor Necrosis Induced by CXCR4 Antagonism

To investigate the pathological mechanism of tumor necrosis induced by the CXCR4 antagonist, we next investigated structures of vessels in the necrotic tumor. Tumor tissues were likely islands interspersed with necrotic tissue in the AMD3100 group (Figure 4a, left). With a higher power magnification, we found that vessel-associated tumor cells survived, whereas those away from the vessels were necrotic (Figure 4a, right). Tumor island-associated vessels expressed CD34 and, thus, were confirmed to be endothelial cells (Figure 4b,c). The histologically unique phenomenon of tumor necrosis associated with disorganized vasculatures was defined as tumor angiogenic inhibition with tumor necrosis (TAITN) (Figure 4a–c). (Later data prompted us to minor-change the definition.) The vessels located at the center of TAITN were CXCR4-positive (Figure 4d). Any TAITN-like structure was not seen in the saline group. In the saline group, CD34-positive tumor vessels did not infiltrate the necrotic area. TAITN significantly occurred in the AMD3100-treated group but was not found in the control tumors (Figure 4e).

These results indicated that the CXCR4 antagonist triggered TAITN in OSCC tumors.

### 3.4. Hypoxia-Inducible Factor-1α Was Induced in Necrotic Tumors under CXCR4 Antagonism

We next hypothesized that the TAITN triggered by CXCR4 antagonism could also induce tumor hypoxia inasmuch as angiogenic inhibition. The major regulator of oxygen tension is hypoxia-inducible factor-1 (HIF-1), a transcription factor complex stabilized under low oxygen tension to mediate cellular responses, which activates and binds to a large number of target genes [32]. The HIF-1 complex is composed of alpha and beta subunits. The alpha subunit of HIF-1 (HIF-1α) is rapidly inactivated and degraded by ubiquitination and subsequent proteasomal pathway, a process that is inhibited under hypoxic conditions [33]. These studies prompted us to ask whether HIF-1α levels could be altered in tumors treated with or without AMD3100. We found that HIF-1α-positive tumor areas tended to be larger in the AMD3100 group than those in the control group (Figure 5a). HIF-1α-positive cells spread in the whole tumor area to the tumor periphery in the CXCR4-inhibited group, whereas HIF-1α-positive cells were limitedly localized in the central area of tumors in the control group (Figure 5b). HIF-1α was increased in cancer cells in the AMD3100-treated mice compared to the untreated mice (Figure 5c,d). The number of HIF-1α-positive cells in the tumors was significantly increased in the AMD3100-treated group compared to the control group (Figure 5d).

These results suggested that intratumoral hypoxia was usually limited in the central area of tumors, whereas CXCR4 antagonism enhanced hypoxia occurring in the whole tumor. 

### 3.5. Tumor Vessels Were Disorganized by CXCR4 Antagonism

We next hypothesized that CXCR4 antagonism-induced tumor necrosis and hypoxia could be mediated by disorganization of vessels, whereby oxygen and nutrition were limited. This hypothesis prompted us to prove a concept of TAITN to be tumor angiogenic inhibition triggered necrosis. We, therefore, asked whether and how CXCR4 antagonism could disrupt structures of blood vessels occupied by viable (non-necrotic) tumor cells. Tumor blood vessels in the AMD3100-treated group were fewer than those in the control (Figure 6a). The tumor blood vessels in the AMD3100-treated group were thinner, shorter, and poorly branched than those in the control (Figure 6b–e, Table 2). The total and average length of blood vessels in the AMD3100 group was significantly shorter than those in the control group (Figure 6c,d). The ratio of blood vessels longer than 100 μm was lesser in the AMD3100-treated group (1.8%) compared to the saline-treated group (6.9%) (Figure 6e and Table 2). The blood vessels shorter than 50 μm were more abundant in the AMD3100 group (86.2%) compared to the control group (76.7%) (Table 2 and Figure 6e).

These data indicated that the CXCR4 antagonist AMD3100 exerted a powerful anti-angiogenic effect that disorganized blood vessels in OSCC tumors—a proof of concept for TAITN.

## 4. Discussion

Our studies revealed that the blood vessels existing in OSCC stroma highly expressed CXCR4 (Figure 1), and CXCR4 antagonism promoted tumor necrosis in OSCC (Figure 3). Therefore, it was conceivable that CXCR4 highly expressed vessels could support tumor survival by supplying nutrients and oxygen. CXCR4 distributed in the blood vessels in the stroma was thought to be involved in tumor progression and survival, consistent with the absence of CXCR4 in the stromal blood vessels of the nontumor area (Figure 2). Consistently, it has been shown that SDF-1/CXCR4 signaling is important in neoangiogenesis as well as infections and immunity [34,35]. Furthermore, the SDF-1/CXCR4 axis plays decisive roles in tumor progression, including the enhancement of migration and invasion, cancer cell–tumor microenvironment interaction, and angiogenesis [27,36,37]. These studies were consistent with our data evidencing the proangiogenic role of CXCR4 in OSCC. Moreover, the participation of the SDF-1/CXCR4 axis in tumor neovascularization through Vascular Endothelial Growth Factor (VEGF)-dependent and -independent mechanisms have been reported [38]. It was also shown that CXCR4 upregulated VEGF expression through the Akt signaling pathway in breast cancer [38]. In addition, CXCR4 was reported to be upregulated under hypoxic conditions in glioblastoma, via HIF-1α and VEGF [39]. Several previous studies have demonstrated that HIF-1α plays a crucial role in the expression and regulation of CXCR4 [33,39,40]. It has been reported that hypoxia, particularly HIF-1α, may promote the expression of CXCR4 and activate the SDF-1α/CXCR4 signaling axis, contributing to tumor cell invasion and metastasis. Although a number of studies have reported HIF-1 to the CXCR4 axis, it has not been investigated whether CXCR4 antagonism could alter HIF-1 expression in tumors. Our study indicated that tumor angiogenesis was inhibited by antagonizing CXCR4, although this anti-angiogenesis approach could result in tumor hypoxia inducing HIF-1α. However, our data indicate that the increases in the HIF-1α-positive area can be an indicator of tumor hypoxia and TAITN.

It has been shown that AMD3100 acts as a specific antagonist that blocks the binding pocket of CXCR4 [41,42]. The first clinical trials with AMD3100 were designed for the treatment of HIV. Notably, an increased number of white blood cells was observed in healthy volunteers in phase I clinical trials [43]. These findings led to the discovery that AMD3100 mobilizes CD34-positive human hematopoietic stem and progenitor cells from the bone marrow to peripheral blood [44]. AMD3100 was finally approved by the U.S. Food and Drug Administration as a mobilizer of hematopoietic CD34-positive cells from the bone marrow to the circulation [45,46]. However, recent studies applied AMD3100 for cancer treatment [47]. The effective CXCR4-related anti-metastatic therapies may require the daily administration of AMD3100 to continuously prevent the migration of pre-metastatic cells to the metastatic site, which may cause chronic leukocytosis [48]. Our study was performed with the same dose and conditions, whereby no adverse effect of AMD3100 was observed.

Our data notably define novel anti-angiogenic pathophysiology, so-called TAITN. CXCR4 antagonism disorganized vessel structures and induced necrosis in the tumor xenograft mouse model (Figure 3 and Figure 4). Tumor necrosis generally begins at the center of tumor tissue due to intratumoral hypoxia and malnutrition. Such hypoxia and malnutrition occur depending on the distance from tumor blood vessels to the tumor, which inhibits neovascularization, keeping up with tumor growth [49,50]. In our model, CXCR4 antagonism induced tumor necrosis along with hemorrhage (Figure 3). The structure of TAITN was particularly found in the CXCR4-antagonized group but not in the control group (Figure 4). We define TAITN as a pivotal indicator of disorganized branching and angiogenesis of tumor blood vessels. TAITN was induced by CXCR4 antagonism and resulted in hypoxia and enhanced necrosis in tumors.

In this study, suppression of CXCR4 inhibited the formation of tumor blood vessels through reducing the thickness and length of tumor blood vessels (Figure 6). In addition, the CXCR4 expression site in tumor blood vessels was limited to a bifurcation or constricted part of the blood vessels (Figure 2). These results suggest that CXCR4 is involved in tumor blood vessel angiogenesis and branching. As a proof of concept of TAITN, the inhibition of CXCR4 caused angiogenic inhibition, whereby the surrounding tumor tissue was induced with hypoxia and became necrotic. TAITN is a novel, distinctive subtype of necrosis inasmuch as marked suppression of tumor vasculature resulted in the retention of an island-like tumor mass confined by inhibited vessels in the necrotic tissue (Figure 7). Hence, these data could open up CXCR4-targeted personalized therapeutics targeting CXCR4-positive tumor blood vessels.

## 5. Conclusions

CXCR4 plays a crucial role in tumor angiogenesis required for OSCC progression, whereby TAITN induced by the CXCR4 antagonist could be an effective anti-angiogenic therapeutic strategy in OSCC treatment.

## Figures and Tables

**Figure 1 cells-08-00761-f001:**
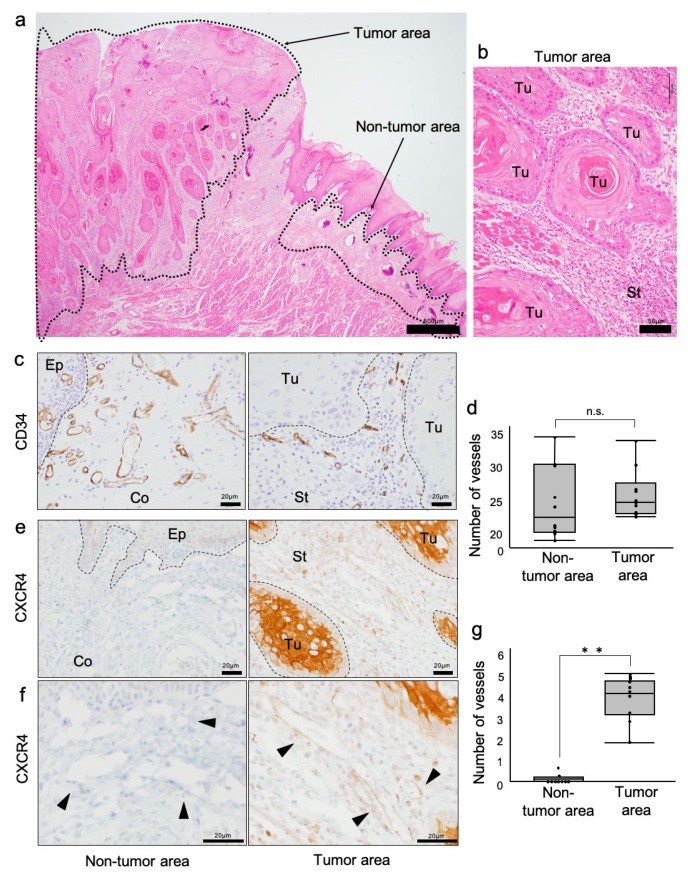
Investigation of CXCR4-positive and CD34-positive vessels in oral squamous cell carcinoma (OSCC) stroma. (**a**) HE staining of an OSCC tissue for the definition of the tumor and nontumor areas. Tumor and nontumor areas are surrounded by dotted lines. (**b**) High-power magnification of tumor area stained with HE. Tu: tumor. St: stroma. (**c**) Immunohistochemistry (IHC) for CD34 in tumor and nontumor areas. Borders between epithelia (Ep), connective tissue (Co), tumor (Tu), and stroma (St) are shown with dotted lines. (**d**) The average number of vessels in the tumor and nontumor areas in a representative OSCC case. *p* = 0.289, n.s., not significant, N = 10 cases. (**e**) IHC for CXCR4 in tumor and nontumor areas. (**f**) High magnification IHC for CXCR4. Arrowheads indicate vessels. CXCR4-positive vessels specifically existed in the tumor area. (**g**) The average number of CXCR4-positive vessels in the tumor and nontumor areas in a representative OSCC case. ** *p* < 0.0001, N = 10 cases.

**Figure 2 cells-08-00761-f002:**
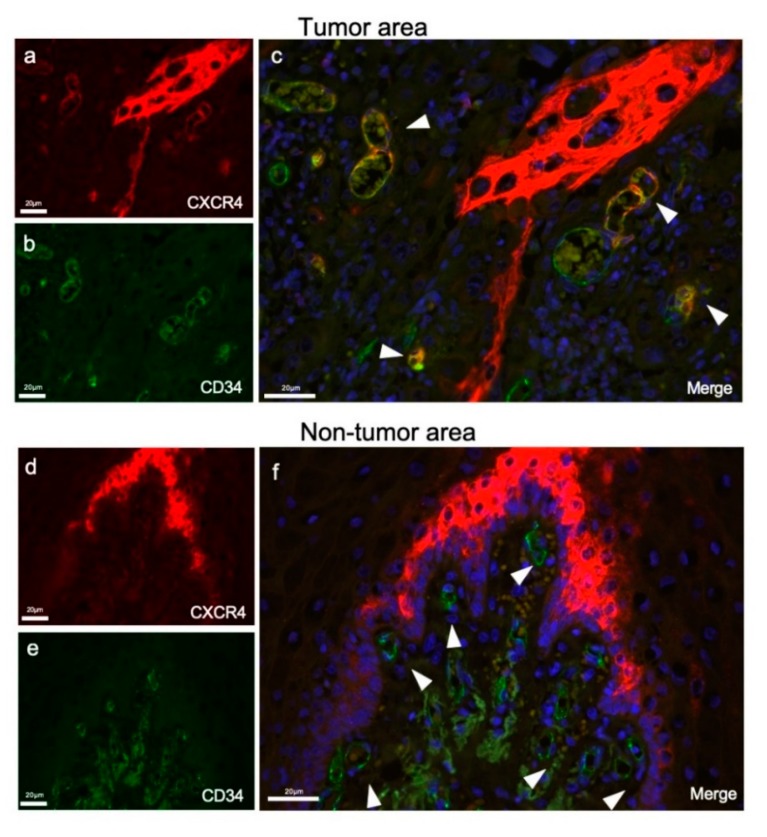
Double-fluorescent IHC for CXCR4 and CD34 in tumor and nontumor areas. (**a**) CXCR4 stained in stromal vessels and tumor cells (red). (**b**) CD34 stained only on vessels in the tumor stroma (green). (**c**) A merged IHC image of CD34 and CXCR4. Nuclei were stained with DAPI. Arrowheads indicate CXCR4/CD34 double-positive tumor vessels in the OSCC stroma. (**d**) CXCR4 stained in the nontumor area (red). (**e**) CD34 stained in the nontumor area (green). (**f**) A merged IHC image of CD34 and CXCR4. Nuclei were stained with DAPI. CD34-positive endothelial cells were all negative for CXCR4 (arrowheads).

**Figure 3 cells-08-00761-f003:**
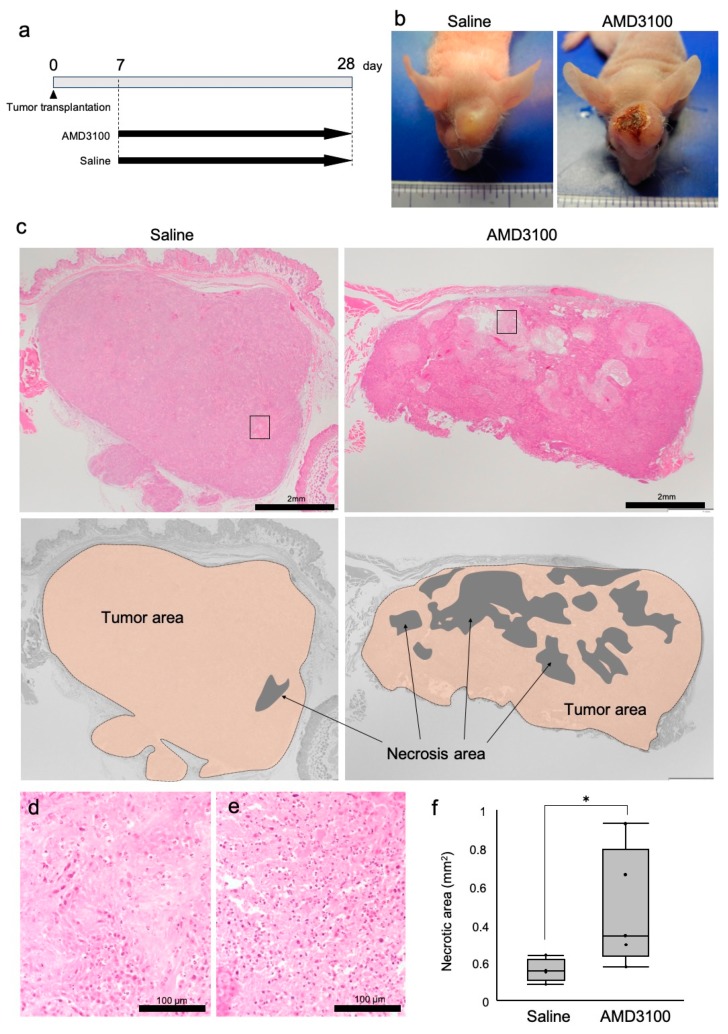
CXCR4 antagonist AMD3100 induced tumor necrosis. Mice were injected with HSC-2 cells subcutaneously in the back and head. Seven days after tumor transplantation, AMD3100 or saline was intraperitoneally administrated every day for 21 d. (**a**) A schematic protocol of tumor injection and AMD3100 administration. (**b**) Representative photographs of subcutaneous tumors formed in mice. Necrosis and bleeding were observed only in the AMD3100 group. (**c**) Definition of necrotic areas. Top, HE staining. The areas closed with rectangles are enlarged in “d” and “e”. Bottom, diagrams defining necrotic areas in each tumor. (**d**,**e**) High-power magnification views of tumor necrotic areas in the saline-treated mouse (**d**) and AMD3100-treated mouse (**e**). The areas are enlarged from Figure 3c. (**f**) Box-whisker-dot plot of tumor necrotic areas with or without AMD3100 administration. * *p* = 0.0416, N = 5 mice.

**Figure 4 cells-08-00761-f004:**
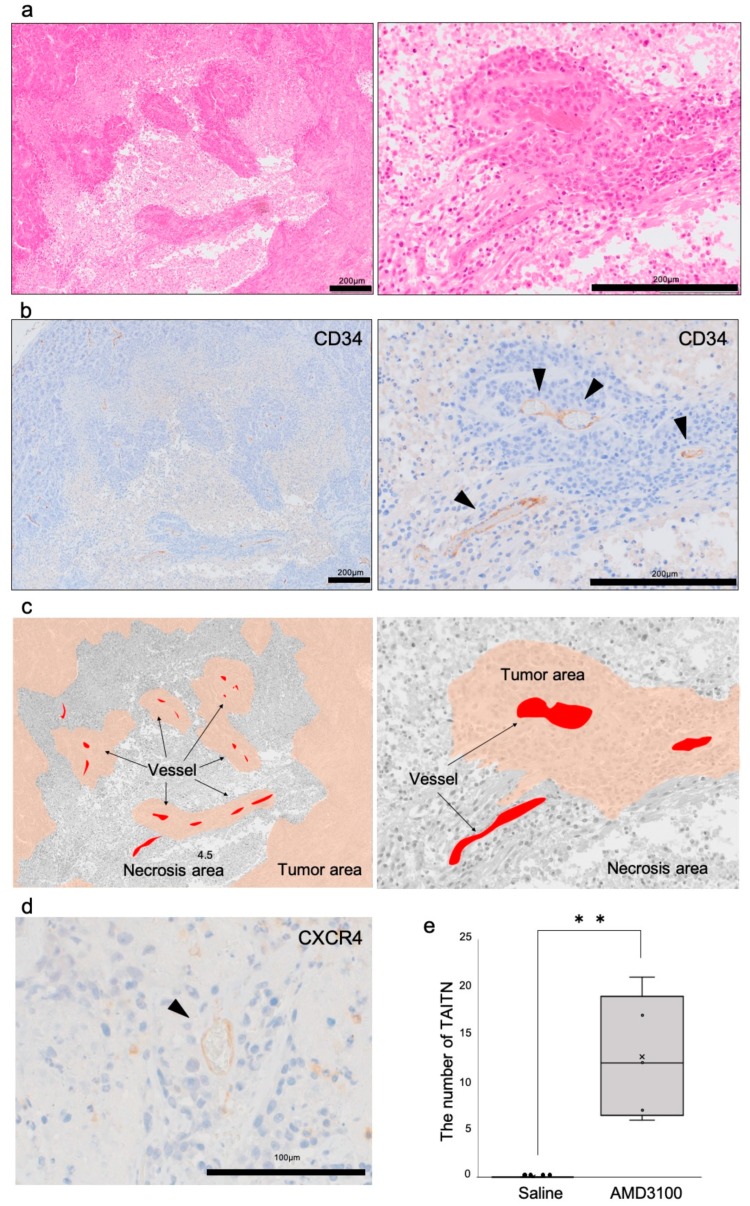
Pathological analysis of tumor angiogenic inhibition triggered necrosis (TAITN) induced by CXCR4 antagonism. (**a**) Representative HE staining of tumor necrotic area in an AMD3100-treated mouse. (**b**) Representative IHC for CD34 in the tumor necrotic area in an AMD3100-treated mouse. (**c**) Illustration of the structure of TAITN found in AMD3100-treated mice. Gray, necrotic area. Red, tumor vessels. Brown, tumor area. Islanded tumor areas are surrounded by necrotic area. The center of each tumor island is occupied by vessels. We defined this unique pathophysiological conception as TAITN. (**d**) IHC for CXCR4 in the TAITN. CXCR4-positive blood vessels were observed at the center of TAITN. (**e**) The number of vessel-tumor islands in the AMD3100 group and saline group. ** *p* = 0.0031, N = 5 mice.

**Figure 5 cells-08-00761-f005:**
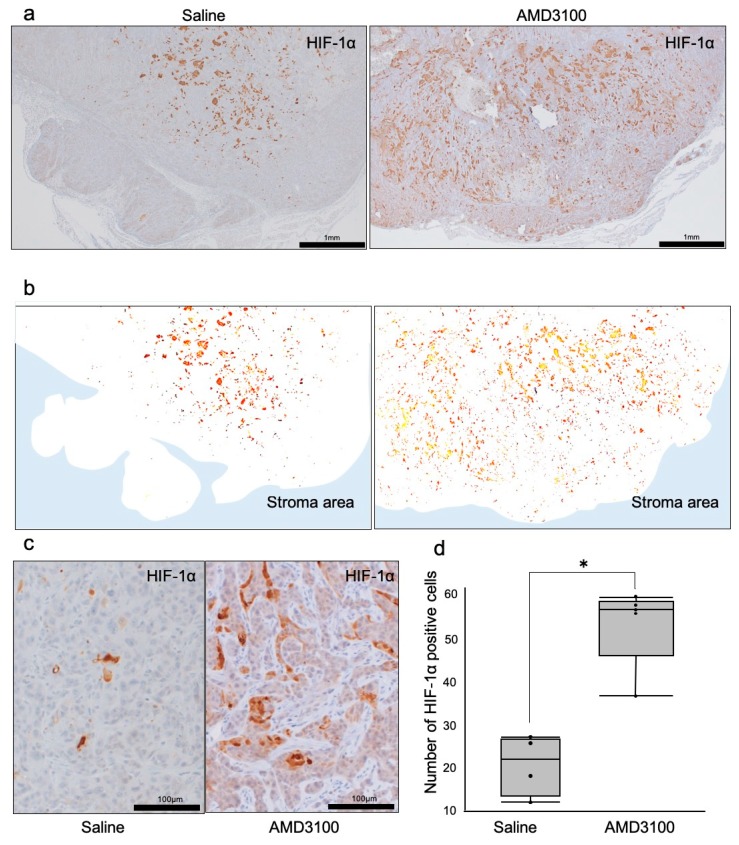
Distribution of hypoxia-inducible factor-1α (HIF-1α) in tumors treated with or without CXCR4 antagonist AMD3100. (**a**) IHC for HIF-1α in the tumor area under low-power magnification. (**b**) Illustration of HIF-1α distribution in tumors. White area, tumor. Blue area, stroma. (**c**) IHC of HIF-1α under high-power magnification. (**d**) The number of HIF-1α-positive tumor cells. * *p* = 0.02, N = 5 mice.

**Figure 6 cells-08-00761-f006:**
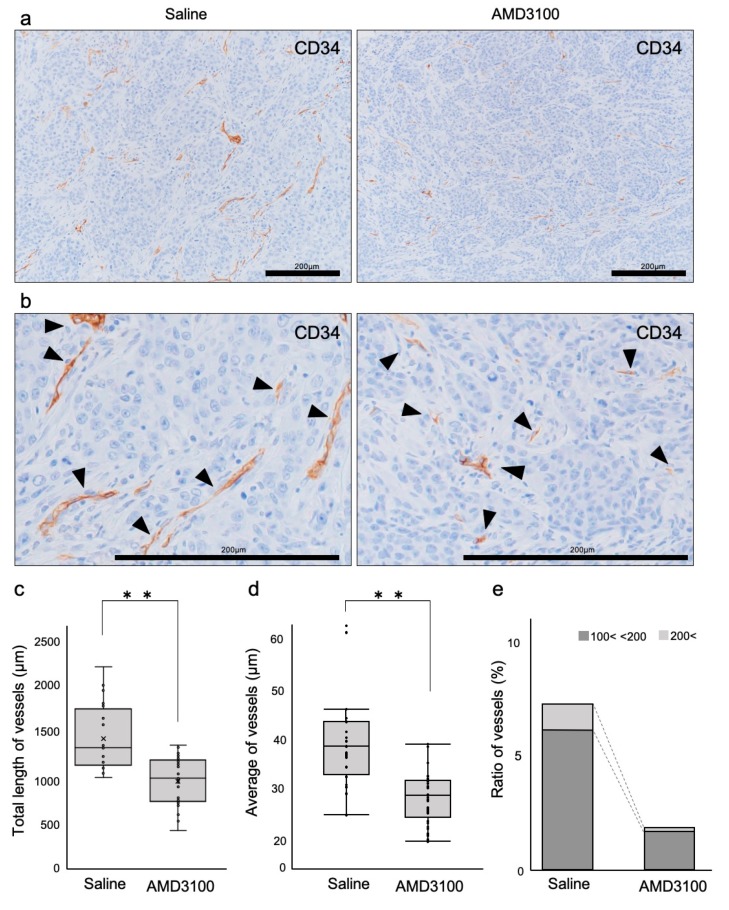
Tumor vessels were disorganized by CXCR4 antagonism. (**a**) Low-power magnification IHC of CD34. (**b**) High-power magnification IHC of CD34. (**c**) The total length of CD34-positive vessels. ** *p* < 0.0001, N = 25. (**d**) The average length of CD34-positive vessels. ** *p* < 0.0001, N = 25. (**e**) Rate of vessels sized 100~200 μm, and >200 μm.

**Figure 7 cells-08-00761-f007:**
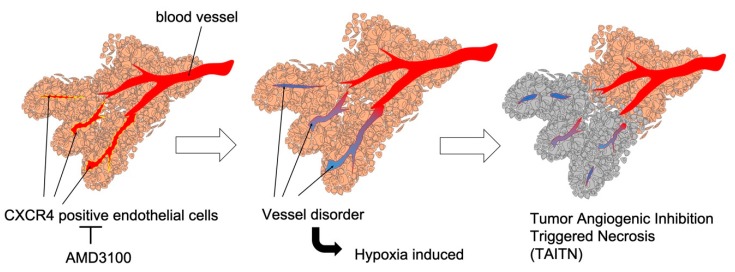
Graphical abstract: CXCR4 antagonism induced tumor necrosis through vessel disorder. Left, a CXCR4 antagonist AMD3100 inhibited CXCR4-positive endothelial cells of tumor vessels. Middle, inhibition of CXCR4-positive endothelial cells induced vessel disorder and caused hypoxia in surrounding tumor cells. Right, necrosis was caused in the surrounding tumor cells through vessel disorder. Taken together, we propose a novel mechanistic concept of “Tumor Angiogenic Inhibition Triggered Necrosis (TAITN)”.

**Table 1 cells-08-00761-t001:** List of patient-derived samples.

Sex.	Age	T	N	M	Stage
Male	78	T2	N2b	M0	IV a
Male	61	T1	N0	M0	I
Female	76	T2	N0	M0	II
Female	39	T2	N0	M0	II
Male	30	T1	N0	M0	I
Male	62	T1	N0	M0	I
Male	52	T2	N0	M0	II
Female	86	T2	N0	M0	II
Male	72	T1	N0	M0	I
Male	76	T1	N0	M0	I

Assessment of the primary tumor (T), assessment of the regional lymph nodes (N), assessment of distant metastasis (M), and Stage criteria follow the TNM classification defined in Malignant Tumors 8th Edition, published by Union International Cancer Control [30] and the detailed criteria of the stages are shown in Appendix A.

**Table 2 cells-08-00761-t002:** Rate of vessels sized <50 μm, 50~100 μm, 100~200 μm, and >200 μm.

	<50	50≤, <100	100≤, <200	200≤
Saline	76.7%	16.4%	5.8%	1.1%
AMD3100	86.2%	11.9%	1.6%	0.2%

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
