# Peer review of "Tumor Angiogenic Inhibition Triggered Necrosis (TAITN) in Oral Cancer"

_cells, 2019, doi:10.3390/cells8070761_

Round 1

Reviewer 1 Report

This is a very interesting paper describing the role of CXCR4 in TAITN of OSCC tumors. The paper is well written and easy to follow, thus may be recommended for publication. I only have two questions.

1. Since authors used a human HSC-2 cell line in present experiment they should show some date whether the mutations of used passage of transplanted cancer cells were stable.

2. Have authors also looked at apoptosis markers in the tumors following AMD3100 treatment?

Author Response

1. Since authors used a human HSC-2 cell line in present experiment they should show some date whether the mutations of used passage of transplanted cancer cells were stable.

HSC-2 was purchased from National Institutes of Biomedical Innovation, Health and Nutrition (Osaka, Japan) on January 14th, 2015, made frozen stocks, and stored in liquid nitrogen. Cells thawed before transplantation and used for experiments (less than 10 passages). As evidence of the maintenance of the cell line, “the transplanted HSC-2 cells formed tumor nest with keratinization and maintained the histological features as well-differentiated SCC. There was neither obvious vascular invasion nor perineural invasion.” From these characteristic findings, we judged that the characteristics of HSC-2 were maintained well.“ We added these sentences in the Results section.

2. Have authors also looked at apoptosis markers in the tumors following AMD3100 treatment?

Morphologically, cellular necrosis was broadly seen, although not apoptosis. To avoid confusion, we edited the sentences as “Morphologically, in the necrotic areas of the two groups, nuclei of tumor cells were lost, cell cytoplasm was swollen, and eosin-stained debris was seen (Fig. 3d), although the concentrated or broken part of the nucleus was barely found, indicating that the main manner of cell death was necrosis in the tumors, although not apoptosis. The necrotic areas were significantly enlarged in the AMD3100 group compared to the control group (Fig. 3e).” Thus, in this study, we have not looked at any apoptosis marker.

We appreciate the reviewer for careful peer-review.

Reviewer 2 Report

Dr. Yoshida and colleagues report on the presence of CXCR4 on CD34+ tumor vasculature in oral squamous cell carcinoma. Using an OSCC xenograft model, the authors demonstrate that a CXCR4 antagonist induced tumor necrosis and HIF1a, and decreased vessel length and vessel numbers. Overall, the manuscript is very rigorous and interesting. A few comments: 

Major comment:

1) The authors propose that TAITN starts with blood vessel changes and leads to tumor necrosis. However, CXCR4 inhibition may be directly inducing tumor apoptosis which then results in decreases vasculature. Please comment on this. Also, although it may be difficult to prove that AMD3100 is primarily working on the vasculature, it would interesting to readers to know whether AMD3100 inhibits/kills OSCC cells in vitro. This may provide some support that the impact of AMD3100 on non-tumor cells 

Minor comments:   

2) Please confirm and clarify that N values in figure legends refer to numbers of mice used for the experiment. (rather than a number of slides, for example) 

3) Figure 1 does not show double positive cells (figure 2 does), so the title of figure 1 should not say "double positive". Also for figure 1C right side, it does not look like any CD34+ vessels are within the Tumor (Tu) region, but rather they are in the stroma region, is this correct? CXCR4 on stromal vasculature is difficult to see (even with arrows) can the authors include an image with increased magnification? Finally for 1D and 1F what is denominator for the vessel # (ie some unit of area or one slide?)?

Overall very interesting. One additional consideration is for the title of the graphical abstract (figure 7). I would suggest adding in some words to make it clear that TAITN is still a concept and was not proven (in my opinion). 

For example "proposed mechanism..." 

Author Response

Major comment:

1) The authors propose that TAITN starts with blood vessel changes and leads to tumor necrosis. However, CXCR4 inhibition may be directly inducing tumor apoptosis which then results in decreases vasculature. Please comment on this. Also, although it may be difficult to prove that AMD3100 is primarily working on the vasculature, it would interesting to readers to know whether AMD3100 inhibits/kills OSCC cells in vitro. This may provide some support that the impact of AMD3100 on non-tumor cells

Despite the fact that OSCC cells express relatively high levels of CXCR4, it has been reported that neither cancer cell proliferation nor survival was inhibited by CXCR4 antagonists. We mentioned this in the Introduction and the Discussion sections with references. Moreover, in the CXCR4 antagnist administration group, the necrosis area was increased in the tumor tissue compared to the control. Histologically, apoptotic cells were barely seen as the very low level as usually found in well-differentiated SCC. These results suggest that necrosis of the tumor tissue was not caused by apoptosis of cancer cells. Therefore, CXCR4 inhibition is considered not to be involved in the induction of apoptosis in cancer cells.

References

·           Almofti, A.; Uchida, D.; Begum, N.M.; Tomizuka, Y.; Iga, H.; Yoshida, H.; Sato, M. The clinicopathological significance of the expression of CXCR4 protein in oral squamous cell carcinoma. Int. J. Oncol. 2004, 25, 65–71.

·           Uchida, D.; Onoue, T.; Kuribayashi, N.; Tomizuka, Y.; Tamatani, T.; Nagai, H.; Miyamoto, Y. Blockade of CXCR4 in oral squamous cell carcinoma inhibits lymph node metastases. Eur. J. Cancer 2011, 47, 452–9.

·           Zlotnik, A. Chemokines in neoplastic progression. Semin. Cancer Biol. 2004, 14, 181–5.

Minor comments:  

2) Please confirm and clarify that N values in figure legends refer to numbers of mice used for the experiment. (rather than a number of slides, for example)

The N values in the figures were denoted as e.g., N = 5 mice.

3) Figure 1 does not show double positive cells (figure 2 does), so the title of figure 1 should not say "double positive". Also for figure 1C right side, it does not look like any CD34+ vessels are within the Tumor (Tu) region, but rather they are in the stroma region, is this correct? CXCR4 on stromal vasculature is difficult to see (even with arrows) can the authors include an image with increased magnification? Finally for 1D and 1F what is denominator for the vessel # (ie some unit of area or one slide?)?

We corrected the title of Figure 1 as “Investigation of CXCR4-positive and CD34-positive vessels in OSCC stroma.” CD34-hi vessels and CXCR4-hi vessels existed in the tumor stromata. According to this editor’s suggestion, we added pictures of high magnification of CXCR4 staining (new Fig. 1f). We counted the number of lumen structures in 5 randomly selected locations of 10 patients (Fig. 1g), as described in the method section.

Overall very interesting. One additional consideration is for the title of the graphical abstract (figure 7). I would suggest adding in some words to make it clear that TAITN is still a concept and was not proven (in my opinion). For example "proposed mechanism...

According to the editor’s suggestion, we edited the description in the graphical abstract- “CXCR4 antagonism induced tumor necrosis through vessel disorder. Left, a CXCR4 antagonist AMD3100 inhibited CXCR4-positive endothelial cells of tumor vessels. Middle, inhibition of CXCR4-positive endothelial cells induced vessel disorder and caused surrounding tumor cells' hypoxia. Right, necrosis was caused in the surrounding tumor cells through vessel disorder. Taken together, we propose the novel mechanistic concept of Tumor Angiogenic Inhibition Triggered Necrosis (TAITN).”

We appreciate the reviewer for careful peer-review.